# Bidirectional Two-Sample, Two-Step Mendelian Randomisation Study Reveals Mediating Role of Gut Microbiota Between Vitamin B Supplementation and Alzheimer’s Disease

**DOI:** 10.3390/nu16223929

**Published:** 2024-11-18

**Authors:** Yu An, Zhaoming Cao, Yage Du, Guangyi Xu, Jingya Wang, Jie Zheng, Yanhui Lu

**Affiliations:** 1Department of Endocrinology, Beijing Chao-Yang Hospital, Capital Medical University, Beijing 100020, China; anyu900222@mail.ccmu.edu.cn; 2School of Nursing, Peking University, Beijing 100191, China; 2311110250@bjmu.edu.cn (Z.C.); yuanfang166@163.com (Y.D.); xuguangyi1998@163.com (G.X.); 2411210127@stu.pku.edu.cn (J.W.); zhengj1_27@163.com (J.Z.)

**Keywords:** gut microbiota, vitamin B, Alzheimer’s disease, Mendelian randomisation

## Abstract

Objectives: Alzheimer’s disease (AD) is a devastating neurodegenerative disorder with a complex aetiology. The aims of this study were to investigate the relationship between vitamin B supplementation and AD risk and to explore the potential mediating effect of the gut microbiota in this relationship. Methods: We employed a Mendelian randomisation analysis to examine the association between different vitamin B supplementation modalities (vitamin B_6_, folic acid, B_12_, and vitamin B complex tablets) and AD risk. Univariate Mendelian randomisation with inverse-variance weighting was used. Additionally, mediation analyses were conducted to identify the potential mediating effects of 119 known bacterial genera. Results: The univariate Mendelian randomisation analyses showed no significant direct associations between individual vitamin B supplements or vitamin B complex tablets and AD risk. However, several gut bacterial genera were significantly associated with AD risk. *Lachnospiraceae* (NK4A136 group), Paraprevotella, Slackia, and Bifidobacterium were associated with reduced AD risk, while *Defluviitaleaceae* (UCG011), *Desulfovibrio*, *Eubacterium ventriosum* group, and *Ruminococcaceae UCG-003* were associated with increased AD risk. The mediation analysis revealed that *Lachnospiraceae* (NK4A136 group), *Defluviitaleaceae* (UCG011), and *Bifidobacterium* fully mediated the causal relationships between vitamin B_12_, B_6_, and B complex supplementation, respectively, and AD risk. Conclusions: This study provides evidence suggesting that certain gut microbiota genera are significantly associated with AD risk and may mediate the relationship between vitamin B supplementation and AD risk. These findings offer new insights into the variable effectiveness of B vitamins in treating neurodegenerative diseases and suggest potential new strategies for AD treatment and prevention.

## 1. Introduction

Alzheimer’s disease (AD) is a progressive neurodegenerative disease with an insidious onset, placing a burden on public health systems worldwide. According to the AD International Report, the number of individuals with AD worldwide is expected to reach 139 million by 2050 [1]. Despite significant advances in the diagnosis and treatment of AD in recent years, a cure remains elusive. Therefore, research is focused on identifying modifiable risk factors before the onset of clinical manifestations to prevent and slow the progression of AD.

B vitamins play an essential role in maintaining neurological function in the brain. Several studies have suggested that vitamin B_12_, folic acid, and vitamin B6 affect cognitive function [2]. Vitamin B_12_ can directly affect neurotransmitter synthesis and myelin formation by interfering with nucleic acid synthesis and methylation. Allen et al. [3] found that vitamin B_12_ deficiency is associated with demyelination, atrophy, and severe polyneuropathy. Several cross-sectional studies have confirmed that vitamin B_12_ is a key molecule in brain development and that deficiency is strongly associated with structural and functional changes in the brain [4,5]. Folate is involved in one-carbon metabolic pathways that influence DNA synthesis and methylation and that maintain normal nervous system function. Many cohort studies have reported that higher folate intake is associated with improved cognitive function and a decreased risk of AD [6,7,8]. Furthermore, a single-blind randomised controlled trial [9] demonstrated that combined folic acid supplementation with vitamin B_12_ improved cognitive dysfunction in older adults. Vitamin B6 is a coenzyme for neurotransmitter synthesis, and it is essential for the synthesis of neurotransmitters such as serotonin, dopamine, norepinephrine, and glycine, which are involved in neural conduction and the protection of neurons. Vitamin B_6_ levels in the central nervous system (CNS) are strongly associated with cognitive function, and a deficiency in this vitamin can exacerbate neurodegenerative diseases. Studies have shown complex interactions between vitamin B6 and other types of B vitamins as well as cognition, brain structure, and functional connectivity in older adults. Some studies have shown that supplementation with B vitamins has potential preventive and therapeutic effects in AD, but other clinical studies have failed to confirm these findings, and further research is needed to clarify the mechanisms of action.

The gut microbiota, a complex ecosystem in the human body, has received increasing attention for its interaction with the nervous system. This microbial community primarily comprises Bacteroidetes and Firmicutes phyla, with Actinobacteria, Proteobacteria, and Verrucomicrobia as minority members [10]. The gut microbiota can produce and affect the absorption of vitamins, particularly B vitamins, which can be synthesised and absorbed in the colon. Recent research suggests that gut bacteria can produce seven of the eight B vitamins [11], and it is estimated that the gut microbiome could contribute up to one-third of the daily reference intake of B vitamins [12,13]. However, the actual bioavailability of these microbiota-produced vitamins varies significantly, as exemplified by vitamin B_12_, which can only be absorbed in the ileum, making colonic production less relevant for host nutrition. The gut microbiota can influence the structure and function of the CNS through multiple pathways and functions. This interaction occurs through various mechanisms, including the direct production of neuroactive compounds, the modulation of vitamin absorption and metabolism, the influence on immune function and inflammation, the production of short-chain fatty acids (SCFAs), and the regulation of intestinal barrier integrity. The relationship between B vitamins and the gut microbiota is bidirectional—while gut bacteria produce B vitamins, these vitamins, in turn, influence microbial composition and function. For instance, vitamin B_6_ supplementation has been shown to affect the abundance of specific bacterial genera, such as *Lachnospiraceae* (NK4A136 group) and Prevotella [10]. Additionally, B vitamins play crucial roles in bacterial metabolism and survival, supporting the fitness of symbiotic species while suppressing the growth of competitive species [11,13]. Emerging evidence suggests that B vitamin metabolism in the gut may influence cognitive function through multiple mechanisms, including the regulation of neurotransmitter synthesis, modulation of neuroinflammation, and maintenance of intestinal barrier integrity. However, research on the causal associations between the gut microbiota, vitamin B intake, and the risk of developing AD is limited.

Observational studies can only reflect the correlation between B vitamin intake, the gut microbiota, and AD [14,15,16] due to the presence of confounding factors and reverse causality, and it is difficult to elucidate the complex underlying causal relationships. Randomised controlled trials are the most reliable method for causal inference, but they are difficult to use directly in disease aetiology analyses owing to the stringent conditions of trial design and implementation and ethical considerations [17]. The primary objective of this study was to examine the potential causal associations among various forms of vitamin B supplementation, specific genera of gut microbiota, and the risk of AD using Mendelian randomisation methodology. Additionally, this study aimed to investigate the intermediary function of the gut microbiota in the relationship between vitamin B supplementation and the development of AD in response to the limitations of observational studies and randomised controlled trials.

## 2. Materials and Methods

### 2.1. Study Design

We used a bidirectional two-sample Mendelian randomisation approach to investigate the causal relationship between different types of vitamin B supplementation and the risk of AD onset. We then conducted causal assessments of the gut microbiota, encompassing 119 species, to identify specific gut bacterial genera that may act as risk or protective factors in the development of AD. Lastly, a two-step Mendelian randomisation strategy was used to examine the potential moderating effects of these gut bacteria on the association between different B vitamin supplementation modalities and the risk of AD. The main purpose of the two-step Mendelian randomisation was to explore whether a mediator could mediate the effect of exposure on outcome and to identify potential pathogenetic mechanisms. Figure 1 illustrates the study design.

### 2.2. Data Sources

Genome-wide association study (GWAS) data on different types of vitamin B supplementation, including vitamin B_6_, vitamin B_12_, and folic acid, individually and as part of multivitamin supplementation, were obtained from the UK Biobank (UKB). For the mediator variables, we used summary statistics of the gut microbiota from the largest meta-analysis of GWASs conducted by the MiBioGen consortium (n = 13,266) [18]. The study included 18,340 individuals from 24 cohorts, most of whom were of European ancestry (n = 13,266), to analyse the microbial composition targeting the variable regions V4, V3–V4, and V1–V2 of the 16S rRNA gene and to classify them using a direct taxonomic classification. Host genetic variation associated with genetic loci linked to the abundance levels of bacterial taxa in the gut bacteria was identified using a microbiota quantitative trait locus (mbQTL) positional analysis. The genera were classified at the lowest level, with 131 genera being identified with a mean abundance greater than 1%, including 12 unknown genera. Therefore, a total of 119 genus-level taxa were included in this study for analysis [18,19]. After excluding UK Biobank participants who had withdrawn consent, we defined cases as any participants who were algorithmically classified as having Alzheimer’s disease (UKB ID 42021, n = 954) and non-cases as any participants who were not (n = 487,331). The analysis was conducted using BOLT-LLM, adjusted for age, sex, genotyping chip, and the first 10 genetic principal components, in the Medical Research Council-Integrative Epidemiology Unit UK Biobank GWAS pipeline. Full methods have been described elsewhere (supplement material 1:MRC IEU UK Biobank GWAS pipeline version 2). Information on all the data used in this study is presented in Table 1.

### 2.3. Genetic Instrument Selection

SNPs used as instrumental variables (IVs) were selected according to the following criteria:(1)Significance: Single-nucleotide polymorphisms (SNPs) associated with each exposure and mediator at the locus-wide significance threshold (*p* < 1.0 × 10^−5^) [19] were used as potential IVs.(2)Independence: European genotypes from the International 1000 Genomes Project (1KGP) were used as a reference panel [20], and the criterion for linkage disequilibrium was set to r^2^ < 0.001, with a window size of 5000 kb. Highly correlated SNPs were excluded to ensure that the included SNPs were mutually independent [21].(3)SNPs with a low allele frequency (≤0.01) were removed.(4)Palindromic sequences: When palindromic SNPs were found, the allele frequency information was used to infer the forward strand allele.(5)SNPs that were significantly associated with Parkinson’s disease, Down’s syndrome, cerebral haemorrhage, cerebral stenosis, and acute cerebral infarction were further excluded to control for the effect of pleiotropy on the findings.(6)Harmonisation: Finally, the SNPs associated with the relative abundance of exposure and mediators were projected onto the pooled GWAS data for AD to extract the corresponding statistical parameters. The data were harmonised according to the statistical parameters with the same loci in the relative abundance of exposure and mediator and AD GWAS results so that the effect values for exposure and outcome corresponded to the same effect allele [22].

### 2.4. Mendelian Randomisation Analysis

A two-step, two-sample Mendelian randomisation analysis was used to explore the associations between different modes of vitamin B supplementation and the risk of developing AD and between 119 different genera of Enterobacteriaceae and the risk of developing AD. Further two-step Mendelian randomisation analyses were performed to assess the possible mediating role of the gut microbiota, which are causally associated with AD, between supplementation with different types of vitamin B and the risk of developing AD.

Five Mendelian randomisation methods were used to estimate causal effects: (1) inverse-variance weighted (IVW), (2) MR–Egger, (3) weighted median (WME), (4) simple mode (SM), and (5) weighted mode (WM) [23,24]. The IVW method assumes that all genetic variants are valid IVs, calculates the causal effect values for individual IVs using the ratio method, and aggregates each estimate into a weighted linear regression to obtain the total effect value. The main difference between the MR–Egger method and the IVW method is that MR–Egger regression takes into account the presence of an intercept term. The WME method exploits the intermediate effects of all available genetic variants and obtains the total effect value by weighting the inverse variance of the correlation of each SNP to the inverse variance of the outcome correlation to obtain an estimate. SM and WM are plurality-based methods. The plurality-based estimation model clusters SNPs with similar causal effects and returns an estimate of the causal effect for the most clustered SNPs. WM weights the effect of each SNP on clustering using the inverse variance of its outcome effect. Because the IVW method is more effective than the other four Mendelian randomisation methods for testing, we used the IVW method as the preferred causal effect estimation method [25]. Furthermore, to better interpret the results, we converted the beta values obtained from the results into odds ratios (ORs) and calculated the 95% confidence interval (CI) of each OR.

Quality control included the use of horizontal multiple validity tests, sensitivity analyses, heterogeneity tests, MR–Steiger directionality tests [26], and F-tests to assess weak instrumental volume bias.
(1)**Multiplicity test**: In this study, we excluded SNPs with significant associations with smoking, physical activity, alcohol consumption, education, obesity, and Parkinson’s disease, based on the existing literature, to rule out the multiplicity of IVs. We further used the Mendelian Randomisation Pleiotropy RESidual Sum and Outlier (MR–PRESSO) method [27] to test whether genetic multiplicity existed between the IVs related to the gut microbiota and AD. Then, we used the MR–PRESSO outlier test to exclude outlier SNPs and to estimate the corrected results.(2)**Sensitivity analysis**: The leave-one-out method was used to examine the effect of each SNP on the results, and the effect of each SNP on the results was assessed by sequentially deleting individual SNPs and calculating the combined effect value of the remaining SNPs [28].(3)**Heterogeneity test**: Cochran’s Q test was used to assess the heterogeneity of the SNPs and to determine possible biases in the estimation of causal effects due to measurement errors of the SNPs caused by different analysis platforms, experimental conditions, and analysis populations.(4)**Directionality test**: The MR–Steiger test was used to assess whether the direction of the assessment results was consistent with the study design.(5)**Weak instrumental variable bias assessment**: The strength of the IVs was assessed by calculating the F-statistic using the following formula:
$$F=\frac{R^{2}\times (N-1-K)}{\left(1-R^{2}\right)\times K}$$where “*N*” *in this formula* represents the sample size, *K* is the number of IVs, and *R*^2^ is the degree to which the IV explains the exposure. An F-statistic > 10 was considered to indicate an absence of bias caused by weak IVs.

### 2.5. Statistical Analysis

Mendelian randomisation analyses were conducted using R programming software (version 4.3.3; R Foundation for Statistical Computing, Vienna, Austria). Two major packages, “TwoSampleMR” and “MRPRESSO”, were used for the estimation of causal effects and the detection of outliers. The results are reported as odds ratios (ORs) with 95% confidence intervals (CIs) per standard deviation.

The mediation proportions were calculated according to the following formula:(β1 × β2)/β0
where β0 represents the total effect obtained from the primary analysis, β1 represents the effect of supplementation with different types of B vitamins on mediators, and β2 represents the effect of mediators on AD.

Standard errors and CIs were calculated using delta methods [29]. If the calculation of the effect size indicated that it was fully mediated, indicating an effect share of 100%, then the mediation effect share did not need to be calculated.

## 3. Results

### 3.1. Bidirectional Two-Sample Mendelian Randomisation Analyses

#### 3.1.1. Causal Effects of Supplementation with Different Types of Vitamin B on AD Traits

The analysis showed no direct causal relationship between vitamin B_6_ supplementation alone, vitamin B_12_ supplementation alone, folic acid supplementation alone, or vitamin B complex supplementation and the risk of developing AD. Figure 2 displays the estimates of the IVW analysis. These findings suggest that the effects of vitamin B on AD risk are mediated by another factor.

#### 3.1.2. Causal Effects of AD Traits on Supplementation with Different Types of Vitamin B

A reverse Mendelian randomisation analysis found that a genetic predisposition to AD traits had no effect on supplementation with different types of vitamin B. Figure 3 displays the estimates of the IVW analysis.

### 3.2. Two-Step Mendelian Randomisation Analyses

#### 3.2.1. Causal Effects of 119 Genus-Level Gut Microbes on AD Traits

According to the selection criteria for IVs, a total of 1572 SNPs were used as IVs for 119 bacterial genera (Appendix A). The IVW analysis identified eight bacterial genera that were significantly associated with AD traits (Appendix A and Figure 4): *Defluviitaleaceae* (UCG011), *Lachnospiraceae* (NK4A136 group), *Paraprevotella*, *Slackia*, *Bifidobacterium*, *Desulfovibrio*, *Ruminococcaceae* (UCG003), and the Eubacterium ventriosum group. Figure 5 displays the estimates of the effect of these eight genus-level gut microbes on AD traits using each of the five Mendelian randomisation methods.

According to the IVW analysis, *Lachnospiraceae* (NK4A136 group) (OR: 0.999, 95% CI: 0.998–0.999, *p* = 0.028, Q = 0.163), *Paraprevotella* (OR: 0.999, 95% CI: 0.999–1.000, *p* = 0.024, Q = 0.064), Slackia (OR: 0.999, 95% CI: 0.998–1.000, *p* = 0.033, Q = 0.064), and *Bifidobacterium* (OR: 0.999, 95% CI: 0.998–1.000, *p* = 0.002, Q = 0.006) are protective factors against AD, and a high relative abundance is associated with a reduced risk of developing AD. *Defluviitaleaceae* (UCG011) (OR: 1.003, 95% CI: 1.000–1.002, *p* = 0.022, Q = 0.058), *Desulfovibrio* (OR: 1.001, 95% CI: 1.000–1.002, *p* = 0.012, Q = 0.023), *Ruminococcaceae* (UCG003) (OR: 1.001, 95% CI: 1.000–1.002, *p* = 0.049, Q = 0.048), and *Eubacterium* (ventriosum group) (OR: 1.001, 95% CI: 1.00–1.001, *p* = 0.036, Q = 0.053) are risk factors for AD, and their elevated abundance increases AD probability.

The MR–PRESSO test did not reveal any potential outliers (global test *p* > 0.05), and sensitivity analyses using the leave-one-out method did not identify any SNP loci in the IVs of the eight bacterial genera that had a strong influence on the results, as shown in Figure 6. The results of the heterogeneity test are shown in Appendix A, and the results of the MR–Steiger directional test were shown to be true, indicating that the correlation between the IVs and gut microbial relative abundance was greater than that between the IVs and AD, confirming that the results of the analyses were in the expected direction. In addition, among the eight causal relationships, the F-statistics of the IVs ranged from 18.95 to 31.36 (Appendix A), excluding the possibility of bias due to weak IVs. The Cochran’s Q test showed that there was no significant heterogeneity in the IVs (Appendix A).

#### 3.2.2. Causal Effects of Different Types of Vitamin B Supplementation on Possible Mediators

We analysed whether supplementation with four different classes of vitamin B with eight genus-level gut bacteria was causally associated with the risk of having AD. The analysis showed that: (1) vitamin B_6_ supplementation was significantly associated with *Defluviitaleaceae* (UCG011) (OR: 0.780, 95% CI: 0.636–0.961; *p* = 0.020); (2) vitamin B_12_ supplementation was significantly associated with *Lachnospiraceae* (NK4A136 group) (OR: 34.890, 95% CI: 2.867–66.916, *p* = 0.031); and (3) vitamin B complex supplementation was significantly associated with *Bifidobacterium* (OR: 7.143, 95% CI: 2.299–11.987, *p* = 0.022) (Figure 7).

### 3.3. Mediation Analysis Results

The results of the mediation analysis indicated that *Defluviitaleaceae* (UCG011) fully mediated the relationship between vitamin B_6_ supplementation alone and AD, *Lachnospiraceae* (NK4A136 group) fully mediated the relationship between vitamin B_12_ supplementation alone and AD, and *Bifidobacterium* fully mediated the relationship between the vitamin B complex supplementation and AD, as detailed in Table 2.

## 4. Discussion

In this study, we investigated the mediating role of the gut microbiota in vitamin B supplementation and AD. Our findings indicate that, while there is no direct causal relationship between vitamin B supplementation and AD risk, specific gut bacterial genera are significant mediators of this relationship. These findings highlight the complex interplay among diet, microbiota, and neurodegenerative disease [30].

Our study findings highlight the potential role of Defluviitaleaceae (UCG011) as a risk factor for AD. This is consistent with recent research suggesting that certain gut microbiota profiles may contribute to the pathogenesis of neurodegenerative diseases [31,32]. The significant inverse causal relationship between vitamin B6 supplementation and Defluviitaleaceae (UCG011) abundance (OR: 0.780, 95% CI: 0.636–0.961; *p* = 0.020) suggests that vitamin B6 may exert protective effects against AD through microbiota modulation. This finding aligns with emerging evidence suggesting that B vitamin-biosynthesising species show distinct population-specific trends in prevalence and abundance [12]. Although the specific mechanisms by which Defluviitaleaceae (UCG011) may influence AD risk are unknown, this bacterial genus could be involved in the production of neurotoxic metabolites or the modulation of inflammatory pathways that are detrimental to cognitive function [33]. Our analysis also revealed a significant inverse causal relationship between vitamin B_6_ supplementation and the abundance of Defluviitaleaceae (UCG011). These findings suggest that vitamin B_6_ may exert protective effects against AD, at least in part by modulating the composition of the gut microbiota. Vitamin B6 plays a crucial role in various metabolic processes, including amino acid metabolism and neurotransmitter synthesis. Vitamin B_6_ supplementation could influence the gut microbial environment, leading to a decrease in the abundance of Defluviitaleaceae (UCG011) [34], which, in turn, may reduce the production of inflammatory or neurotoxic metabolites, thereby mitigating the risk of AD [35]. The interplay between vitamin B_6_ and the gut microbiota underscores the potential of nutritional interventions in the management of neurodegenerative diseases. By modulating the composition of the gut microbiota, vitamin B_6_ supplementation may offer a novel approach to attenuating systemic inflammation and neuroinflammation, which are key factors in AD pathogenesis [36]. This finding aligns with the growing body of evidence supporting the role of the gut–brain axis in neurodegenerative disorders [37].

Our analysis revealed that *Lachnospiraceae* (NK4A136 group) fully mediates the relationship between vitamin B12 supplementation and AD risk (OR: 34.890, 95% CI: 2.867–66.916, *p* = 0.031). This finding is particularly significant in light of recent research showing that B vitamin-biosynthesising bacteria, particularly Lachnospiraceae, demonstrate population-specific variations in prevalence and metabolic capacity. The mechanistic pathway likely involves intrinsic factor, a protein that is secreted by the stomach and is essential for the absorption of vitamin B_12_ in the ileum [38]. Additionally, the transport and cellular uptake of vitamin B_12_ require the presence of transcobalamin, a protein that binds to vitamin B_12_ and facilitates its entry into cells [39]. Deficiencies or impairments in any of these steps could hinder the effectiveness of vitamin B_12_ supplementation in reducing the risk of AD. The relationship between vitamin B_12_ and AD pathogenesis is multifaceted and may involve various mechanisms. Although vitamin B_12_ deficiency has been associated with increased levels of homocysteine, which is a risk factor for AD [40], the exact role of homocysteine in AD development remains unclear [41]. Furthermore, vitamin B_12_ is involved in the synthesis of myelin, a crucial component of nerve cell insulation [42]. Although vitamin B_12_ deficiency can lead to demyelination and neurological dysfunction, the specific impact of this process on the risk of AD requires further investigation. This study revealed that the gut microbiota, particularly *Lachnospiraceae* (NK4A136 group), may mediate the relationship between vitamin B_12_ supplementation and AD risk. *Lachnospiraceae* (NK4A136 group) has been identified as a key member of the gut microbiota involved in the production of short-chain fatty acids (SCFAs), such as butyrate [43]. SCFAs have been shown to exert neuroprotective effects by modulating inflammation, enhancing BDNF expression, and improving blood–brain barrier integrity [44,45]. Additionally, *Lachnospiraceae* (NK4A136 group) may indirectly influence AD pathogenesis by regulating intestinal permeability and modulating the immune system [46]. Vitamin B_12_ supplementation alters the composition and function of the gut microbiota, particularly the abundance of *Lachnospiraceae* (NK4A136 group). This alteration may lead to an increased production of SCFAs and other neuroactive metabolites, which, in turn, exert protective effects on the brain and reduce the risk of AD. Furthermore, the modulation of the gut microbiota by vitamin B_12_ supplementation may attenuate systemic inflammation and improve gut barrier function, thereby mitigating the potential contribution of these factors to the development of AD.

The lack of direct effects of vitamin B complex supplementation on the risk of AD may be attributable to several factors. First, the absorption and utilisation of vitamin B in the human body are influenced by various factors such as age, genetic variation, and gastrointestinal conditions [47]. Second, the protective effects of vitamin B on the brain may depend on the timing of the supplementation. Previous studies suggest that vitamin B supplementation may be more effective in the early stages of cognitive decline than in advanced AD [48]. Third, the complex aetiology of AD involves multiple pathways, including amyloid-β accumulation, tau hyperphosphorylation, and neuroinflammation. Therefore, vitamin B may have limited effects on these pathways, especially in the absence of other synergistic factors.

Our study found that Bifidobacterium fully mediates the relationship between vitamin B complex supplementation and AD (OR: 7.143, 95% CI: 2.299–11.987, *p* = 0.022). Recent research by LeBlanc et al. [49] demonstrated that Bifidobacterium species, particularly through their B vitamin production capabilities, can significantly influence inflammatory responses and neurological function. This finding is particularly relevant given that Bifidobacterium species can both synthesise vitamin B and promote its absorption, produce neuroactive compounds such as γ-aminobutyric acid (GABA), and maintain intestinal barrier integrity, crucial for gut–brain axis function [50,51,52,53,54,55]. These functions are highly relevant to brain health, as they help to maintain a balanced immune response, prevent the translocation of harmful substances from the gut to the brain, and regulate neurotransmitter signalling. Furthermore, Bifidobacterium synthesises and promotes vitamin B absorption in the gut [56]. This suggests a bidirectional relationship between B vitamins and Bifidobacterium, in which B vitamins support the growth of Bifidobacterium, and, in turn, Bifidobacterium enhances the bioavailability and utilisation of vitamin B. In the context of AD, this interaction may be particularly important because Bifidobacterium could potentially amplify the neuroprotective effects of B vitamins by optimising their absorption in, and delivery to, the brain. Moreover, Bifidobacterium may exert neuroprotective effects through the modulation of the gut–brain axis. The gut–brain axis is a complex communication network that involves the nervous, endocrine, and immune systems, allowing for bidirectional signalling between the gut and the brain [57]. Bifidobacterium influences the gut–brain axis by regulating the production of neurotransmitters, such as serotonin and GABA, and by modulating the expression of BDNF [58].

This study had some limitations. The reliance of this study on Mendelian randomisation introduces limitations related to the assumptions of the method, such as the exclusion of the pleiotropic effects of genetic variants [22]. Additionally, the generalisability of these findings may be limited to populations similar to those in the UK Biobank, from which the data were derived. Further research should explore the longitudinal impact of vitamin B supplementation on the gut microbiota and AD risk across different populations and dietary backgrounds. The mechanisms through which these bacterial genera influence brain health should be investigated using experimental studies, thereby potentially guiding the development of probiotic treatments for AD. This study enhances our understanding of the indirect pathways through which vitamin B supplementation may influence the risk of AD, mediated by the gut microbiota. This underscores the potential of integrating dietary management with microbiota-targeted therapies to prevent or mitigate the progression of AD.

## 5. Conclusions

In conclusion, this study revealed the potential role of the gut microbiota in mediating the link between vitamin B supplementation and a reduced risk of AD, thereby providing new directions for future research and clinical practice.

## Figures and Tables

**Figure 1 nutrients-16-03929-f001:**
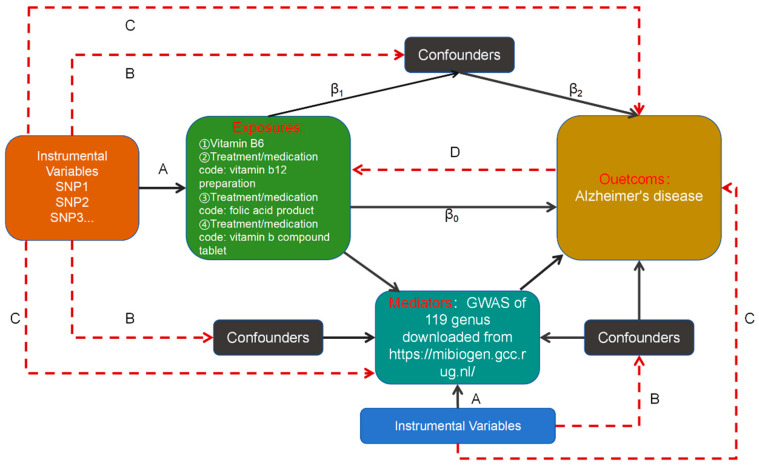
Analytical model and three hypotheses based on Mendelian randomisation (MR) analysis used to study the causal relationship between different modalities of vitamin B supplementation and Alzheimer’s disease and the moderating role of gut bacteria. SNP: single-nucleotide polymorphism. A. The instrumental variables (IVs) must be significantly connected to the exposure. B. The IVs cannot be connected to any known confounders that could alter the association between an exposure and an outcome. C. The IVs must be unrelated to the outcomes and may only affect the outcomes through their effects on the exposure. D: Reverse causation from the outcome to exposure. This figure presents a diagram of our MR study. The dashed lines indicate irrelevance, and the solid lines indicate relevance. In general, we had already determined the causal effect of exposure on the outcome (assuming that it was β0) before performing the two-step MR analysis. Step (1): Significant SNPs were identified in the exposed GWAS findings, SNPs with cascade imbalances were removed, and then the remaining SNP information was extracted from the GWAS findings of the mediator variable; here, it was necessary to make sure that the remaining SNPs could not be directly correlated with the confounders or the mediator variable. Finally, we could calculate the causal effect of exposure on the mediator variable (assuming that it was β1). Step (2): Significant SNPs were identified in the GWAS results of the mediator variable, SNPs with cascade imbalances were removed, and then, the remaining SNP information was extracted from the GWAS results of the outcome; here, it was necessary to ensure that the remaining SNPs could not be directly related to the confounders or the outcome. Finally, we could calculate the causal effect of the mediating variable on the outcome (assuming that it was β2).

**Figure 2 nutrients-16-03929-f002:**
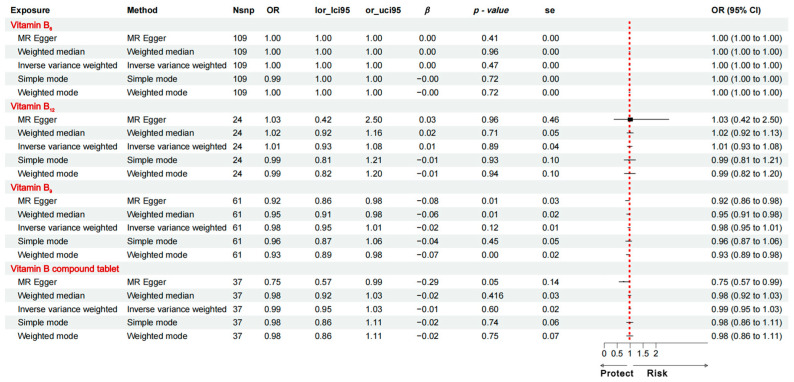
Mendelian randomisation estimates derived from the IVW method to assess the causal effect of vitamin B_6_, vitamin B_12_, vitamin B_9_, and vitamin B complex supplements on AD-related traits. OR: odds ratio; CI: confidence interval.

**Figure 3 nutrients-16-03929-f003:**
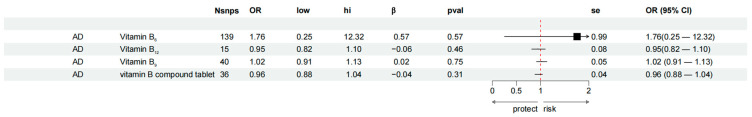
Mendelian randomisation estimates derived from the IVW method to assess the causal effect of AD-related traits on vitamin B_6_, vitamin B_12_, vitamin B_9_, and vitamin B complex supplements. OR: odds ratio; CI: confidence interval.

**Figure 4 nutrients-16-03929-f004:**
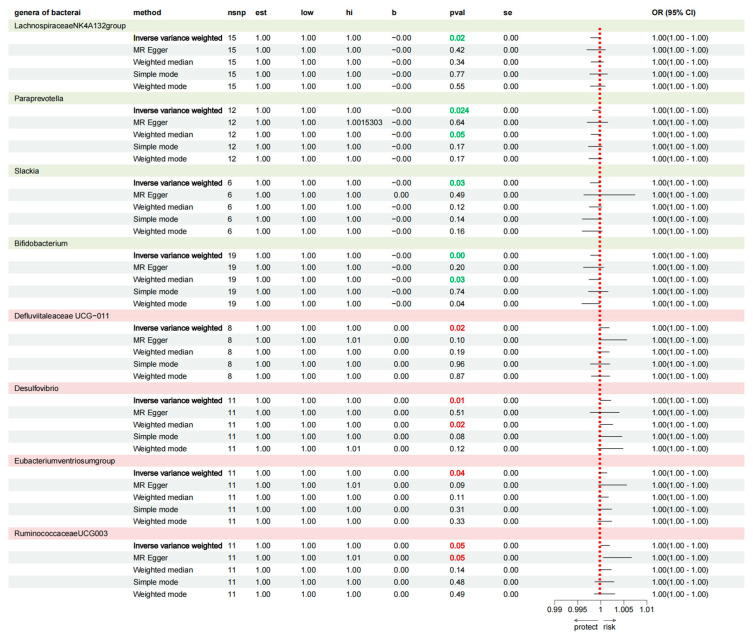
Estimates of the five methods of eight gut microbe genera on Alzheimer’s disease traits. β-value > 1 and *p*-value < 0.05 are marked in red to indicate that this type of gut microbe is a risk factor for the occurrence of AD. β-value < 1 and *p*-value < 0.05 are marked in green to indicate that this type of gut microbe is a protective factor for the occurrence of AD.

**Figure 5 nutrients-16-03929-f005:**
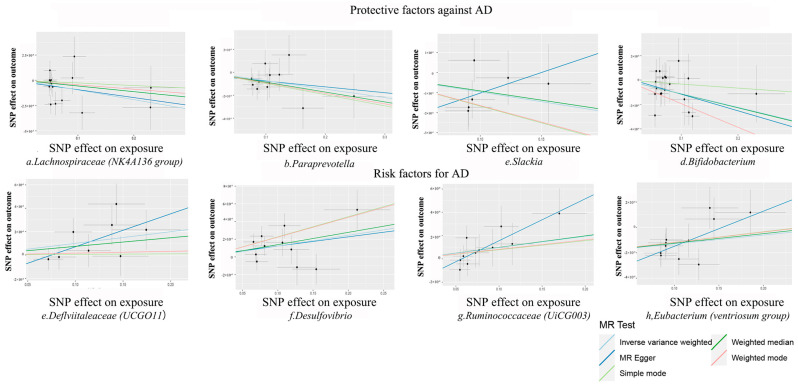
Scatter plots of the causal association between gut microbiota and Alzheimer’s disease. SNP: single-nucleotide polymorphism.

**Figure 6 nutrients-16-03929-f006:**
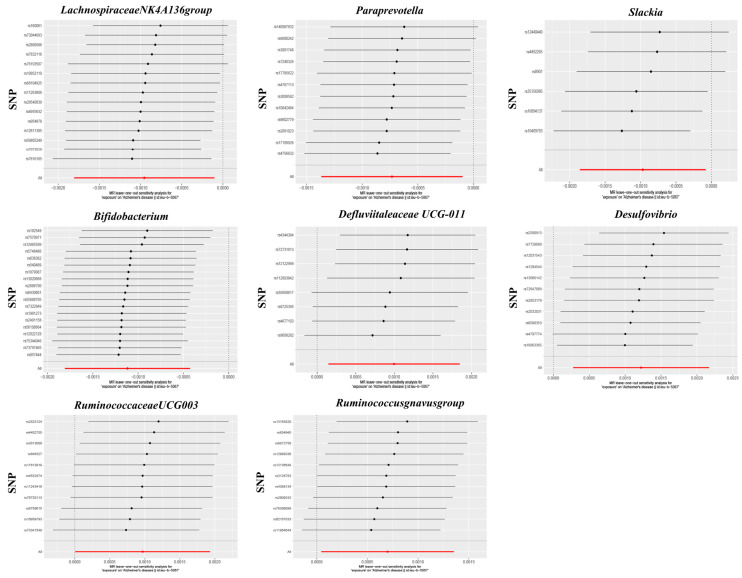
Leave-one-out plots for the causal association between gut microbiota and AD.

**Figure 7 nutrients-16-03929-f007:**
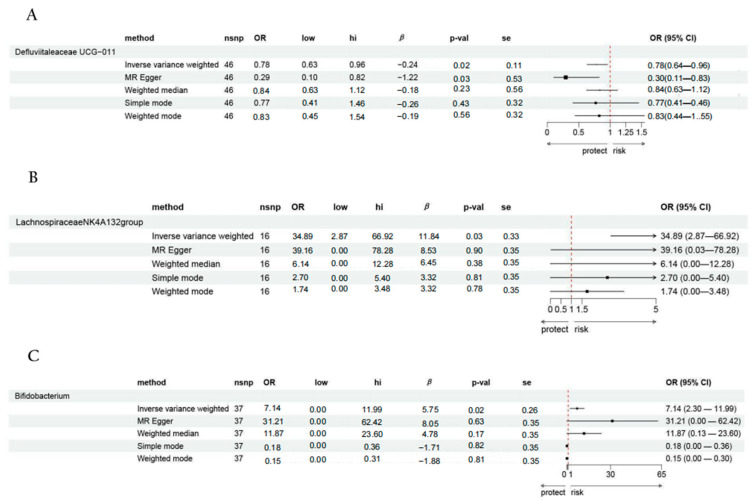
Causal effects of different types of vitamin B supplementation on possible mediators. (**A**) Vitamin B_6_ supplementation*–Defluviitaleaceae* UCG-011–AD supplementation. (**B**) Vitamin B_12_ supplementation–*Lachnospiraceae* NK4A136 group–AD. (**C**) Multivitamin B supplementation–*Bifidobacterium*–AD.

**Table 1 nutrients-16-03929-t001:** Information on GWAS summary statistics for exposure and outcomes in the TSMR.

Type	Variable	Sample Size	Population	Data Download Address
Exposures	Vitamin B_12_	462,933	European	https://gwas.mrcieu.ac.uk/datasets/ukb-b-18819/ (accessed on 5 April 2024)
	Vitamin B_6_	64,979	European	https://gwas.mrcieu.ac.uk/datasets/ukb-b-7864/ (accessed on 5 April 2024)
	Folic acid product	462,933	European	https://gwas.mrcieu.ac.uk/datasets/ukb-b-288/ (accessed on 5 April 2024)
	Vitamin B compound tablet	462,933	European	https://gwas.mrcieu.ac.uk/datasets/ukb-b-2669/ (accessed on 5 April 2024)
Mediators	Gut bacteria	18,340	European	https://mibiogen.gcc.rug.nl/ (accessed on 5 April 2024)
Outcomes	Alzheimer’s disease	488,285	European	https://gwas.mrcieu.ac.uk/datasets/ieu-b-5067/ (accessed on 5 April 2024)

**Table 2 nutrients-16-03929-t002:** Mediator effect analysis.

No.	Exposure–Mediator–Outcome	Mediator Effect
1	Vitamin B_6_ supplementation–*Defluviitaleaceae* UCG-011–AD supplementation	−2.45 × 10^−4^
2	Vitamin B_12_ supplementation–*Lachnospiraceae NK4A136 group*–AD	−1.13 × 10^−2^
3	Multivitamin B supplementation–*Bifidobacterium*–AD	−6.43 × 10^−3^

## Data Availability

The authors confirm that the data supporting the findings of this study are available within the article [and/or] its Appendix A.

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
