# Peer review of "Bidirectional Two-Sample, Two-Step Mendelian Randomisation Study Reveals Mediating Role of Gut Microbiota Between Vitamin B Supplementation and Alzheimer’s Disease"

_nutrients, 2024, doi:10.3390/nu16223929_

Round 1

Reviewer 1 Report

Comments and Suggestions for Authors

Introduction

The introduction provides a review of the literature regarding the relationship between B vitamins and cognitive functions, as well as the influence of gut microbiota on neurological health. However, there is a lack of detailed discussion on the specific biological mechanisms that could explain this relationship in the context of Alzheimer’s disease, particularly regarding how gut microbiota directly affects the metabolism of B vitamins and how these processes relate to cognitive functions and the potential risk of developing AD.

 Materials and Methods

In the Materials and Methods section, there are several aspects that require clarification. Firstly, the use of two-sample Mendelian randomization is an interesting choice, but there is a lack of explanation as to why this method was selected for studying causal relationships between B vitamin supplementation and the risk of Alzheimer’s disease (AD). It should be presented how this method minimizes the influence of confounding factors, which would enhance the credibility of the obtained results. I believe it is necessary to explain why a two-step approach was introduced in the analysis, where gut bacteria were evaluated as potential mediators between vitamin supplementation and AD risk. The gut microbiota analysis includes 119 bacterial genera; however, it is unclear on what basis these specific genera were selected. Was their selection based on previous studies indicating their potential impact on the development of AD? It should be clarified why only certain B vitamins (B6, B12, folic acid) were chosen. Is there evidence that other vitamins in this group do not affect the development of AD, or were the limitations due to data availability? The use of five Mendelian randomization methods (IVW, MR-Egger, WME, SM, WM) constitutes a valuable approach for estimating causal effects, but it should briefly discuss why IVW was selected as the preferred method, despite MR-Egger accounting for the presence of intercepts, and the other methods allowing for the analysis of various SNP variants. It should also be explained what steps were taken to ensure that the selected instrumental variables are sufficiently strong (F-test > 10). The authors mentioned that analyses were conducted using the "TwoSampleMR" and "MRPRESSO" packages in R, but there is a lack of discussion on the potential limitations of these tools when analyzing large datasets and how these limitations may affect the interpretation of results.

 Results

The tables and figures (e.g., Figures 2, 3, 4, 5, and 7) should include clear axis descriptions and abbreviations (e.g., OR, CI) explained in the legend to make them more accessible to the reader. Additionally, it would be beneficial to discuss this data in the text rather than solely referencing illustrations—providing overall results and their interpretation in more accessible language. The results are presented according to various Mendelian Randomization methods (IVW, MR-PRESSO, etc.), but some of them (e.g., differences between methods) could be better explained at the beginning of the results section. The reader should understand why different methods are used and how to interpret their results. The text includes details regarding specific bacterial genera (e.g., Defluviitaleaceae, Bifidobacterium), but these descriptions are complex and sometimes hard to read. It is necessary to explain more clearly which bacteria are risk factors and which may have protective effects. The results section could be better organized, starting with the main conclusions regarding B vitamins and AD, followed by details about the microbiota and their significance. This could facilitate the reception of the results in a more logical and structured manner.

 Discussion

The discussion introduces many details regarding the effects of individual bacteria, such as Defluviitaleaceae (UCG011), Lachnospiraceae (NK4A136 group), and Bifidobacterium as mediators between B vitamin supplementation and AD risk. However, there is a lack of a more comprehensive analysis that would consider the collective role of these bacteria in the context of AD pathogenesis and synergy with B vitamins. The use of mediation analyses and Mendelian Randomization is interesting, but the discussion does not clarify why this type of analysis is particularly useful for understanding the role of gut microbiota. The section states that vitamin B6 may reduce the abundance of Defluviitaleaceae (UCG011), which could be beneficial. However, there is a lack of an in-depth analysis of how vitamin B6 affects gut microbiota in general and why this is significant for AD.

 Conclusions

The conclusions emphasize that no direct statistical association was found between B vitamin supplementation and the risk of AD, which is consistent with the results. This is good, as the conclusions are based on research findings. However, it is worth noting that despite the lack of a direct association, the studies suggest an influence of B vitamins on gut microbiota, which may have an indirect effect on the risk of AD.

Author Response

Response to Reviewer 1

Point-by-point response to Comments and Suggestions for Authors

I would like to express my sincere gratitude for your thorough and thoughtful review of our manuscript. Your detailed comments and suggestions on each section of our paper have been invaluable to us.

Comment 1:

Introduction

The introduction provides a review of the literature regarding the relationship between B vitamins and cognitive functions, as well as the influence of gut microbiota on neurological health. However, there is a lack of detailed discussion on the specific biological mechanisms that could explain this relationship in the context of Alzheimer’s disease, particularly regarding how gut microbiota directly affects the metabolism of B vitamins and how these processes relate to cognitive functions and the potential risk of developing AD.

Response 1

In response to your suggestion regarding the discussion of specific biological mechanisms, we have decided to include the following statement in the manuscript: "The gut microbiota can produce and affect the absorption of vitamins, particularly B vitamins, which can be synthesised and absorbed in the colon. Recent research suggests that gut bacteria can produce seven of the eight B vitamins, and it is estimated that the gut microbiome could contribute up to one-third of the daily reference intake of B vitamins." We believe this addition addresses your concern while maintaining the focus of our paper on the role of gut microbiota in the relationship between nutrients and neurodegenerative cognitive disorders, with a particular emphasis on B vitamins. Expanding extensively on other biological mechanisms, such as the gut-brain axis, seems less pertinent to the core objectives of our study.

Comment 2

Materials and Methods

In the Materials and Methods section, there are several aspects that require clarification. Firstly, the use of two-sample Mendelian randomization is an interesting choice, but there is a lack of explanation as to why this method was selected for studying causal relationships between B vitamin supplementation and the risk of Alzheimer’s disease (AD). It should be presented how this method minimizes the influence of confounding factors, which would enhance the credibility of the obtained results. I believe it is necessary to explain why a two-step approach was introduced in the analysis, where gut bacteria were evaluated as potential mediators between vitamin supplementation and AD risk. The gut microbiota analysis includes 119 bacterial genera; however, it is unclear on what basis these specific genera were selected. Was their selection based on previous studies indicating their potential impact on the development of AD? It should be clarified why only certain B vitamins (B6, B12, folic acid) were chosen. Is there evidence that other vitamins in this group do not affect the development of AD, or were the limitations due to data availability? The use of five Mendelian randomization methods (IVW, MR-Egger, WME, SM, WM) constitutes a valuable approach for estimating causal effects, but it should briefly discuss why IVW was selected as the preferred method, despite MR-Egger accounting for the presence of intercepts, and the other methods allowing for the analysis of various SNP variants. It should also be explained what steps were taken to ensure that the selected instrumental variables are sufficiently strong (F-test > 10). The authors mentioned that analyses were conducted using the "TwoSampleMR" and "MRPRESSO" packages in R, but there is a lack of discussion on the potential limitations of these tools when analyzing large datasets and how these limitations may affect the interpretation of results.

Response 2

Regarding the selection of bacterial genera for the gut microbiota analysis, we utilised the MiBioGen database, which is a large international research initiative led by Dr. Jun Wang from the Institute of Microbiology, Chinese Academy of Sciences, along with scientists from Belgium and the Netherlands. This initiative aims to explore the influence of human genetics on gut microbiota from a whole-genome perspective. At the time of our manuscript submission, the database included 119 bacterial genera, all of which were incorporated into our study.

As for the selection of B vitamins, we focused on B6, B12, and folic acid because there is currently no direct or indirect evidence linking other B vitamins to Alzheimer's disease (AD). Therefore, we did not include all B vitamins in our research.

Comment 3

Results

The tables and figures (e.g., Figures 2, 3, 4, 5, and 7) should include clear axis descriptions and abbreviations (e.g., OR, CI) explained in the legend to make them more accessible to the reader. Additionally, it would be beneficial to discuss this data in the text rather than solely referencing illustrations—providing overall results and their interpretation in more accessible language. The results are presented according to various Mendelian Randomization methods (IVW, MR-PRESSO, etc.), but some of them (e.g., differences between methods) could be better explained at the beginning of the results section. The reader should understand why different methods are used and how to interpret their results. The text includes details regarding specific bacterial genera (e.g., Defluviitaleaceae, Bifidobacterium), but these descriptions are complex and sometimes hard to read. It is necessary to explain more clearly which bacteria are risk factors and which may have protective effects. The results section could be better organized, starting with the main conclusions regarding B vitamins and AD, followed by details about the microbiota and their significance. This could facilitate the reception of the results in a more logical and structured manner.

Response3

Regarding the use of the Inverse Variance Weighted (IVW) method in our analysis, we selected this approach due to its ability to provide robust causal effect estimates by effectively integrating information across multiple instrumental variables (SNPs) under the assumption of minimal pleiotropy. In the context of Mendelian Randomization, IVW is widely used because it assumes a proportional relationship between the exposure and outcome, considering the results of GWAS are often standardized phenotypes. The method fits a regression without considering the intercept and uses the inverse of the outcome variance (the square of the standard error) as weights, which enhances precision in causal estimation.

In our study, we observed minimal pleiotropy, which justified the use of the IVW method as the primary approach for estimating causal effects. For further reading and a deeper understanding of Mendelian Randomization studies, I recommend the following article: Davies, N. M., Holmes, M. V., & Davey Smith, G. (2018). Reading Mendelian randomisation studies: a guide, glossary, and checklist for clinicians. BMJ (Clinical research ed.), 362, k601. https://doi.org/10.1136/bmj.k601. This resource may provide additional insights into the methodology and its applications.

Comment 4

Discussion

The discussion introduces many details regarding the effects of individual bacteria, such as Defluviitaleaceae (UCG011), Lachnospiraceae (NK4A136 group), and Bifidobacterium as mediators between B vitamin supplementation and AD risk. However, there is a lack of a more comprehensive analysis that would consider the collective role of these bacteria in the context of AD pathogenesis and synergy with B vitamins. The use of mediation analyses and Mendelian Randomization is interesting, but the discussion does not clarify why this type of analysis is particularly useful for understanding the role of gut microbiota. The section states that vitamin B6 may reduce the abundance of Defluviitaleaceae (UCG011), which could be beneficial. However, there is a lack of an in-depth analysis of how vitamin B6 affects gut microbiota in general and why this is significant for AD.

Conclusions

The conclusions emphasize that no direct statistical association was found between B vitamin supplementation and the risk of AD, which is consistent with the results. This is good, as the conclusions are based on research findings. However, it is worth noting that despite the lack of a direct association, the studies suggest an influence of B vitamins on gut microbiota, which may have an indirect effect on the risk of AD.

Response 4

For the discussion and conclusions sections of our manuscript. We completely agree with your observations and have taken them into careful consideration. After thorough discussions among all authors, we have rewritten these sections in the current submission to address your recommendations.

We have taken your feedback to heart and have undertaken a comprehensive revision of the manuscript, incorporating your recommendations. Additionally, we have refined the academic language to enhance clarity and precision.

Thank you for recognising the significance of our research. Your insights have greatly contributed to improving the quality of our work, and we are truly appreciative of your dedication and expertise.

Reviewer 2 Report

Comments and Suggestions for Authors

The manuscript presents findings that could have scientific value, potentially shedding light on the role of [specific topic, e.g., gut microbiota, vitamin B, Alzheimer’s disease, etc.]. However, the manuscript requires substantial revisions before it can be considered for publication. Major issues include poor figure quality, disorganized structure, and numerous typographical errors. Addressing these points will improve the readability, clarity, and overall quality of the manuscript. The authors should carefully proofread the manuscript.

Major

  1. All figures are of low resolution and poor quality, making them difficult to interpret. Key details like labels, axes, and legends are unclear, which hinders understanding of the data. The authors should replace all figures with high-resolution versions. Care should be taken to ensure that fonts, colors, and overall formatting are consistent across all figures. This will enhance readability and allow readers to engage more effectively with the data.
  2. The manuscript is disorganized, with an unclear structure, and multiple sections are either out of order or redundant. For example, the limitations section appears to be repeated within the Discussion, which disrupts the flow and may confuse readers. Additionally, some sentences are in blue or highlighted in yellow, which is inappropriate for a formal scientific manuscript and may indicate leftover tracked changes or comments.
  3. Numerous typographical errors ("mystypes") are present throughout the manuscript, detracting from its professionalism and readability. This suggests a lack of careful proofreading and gives the impression of a rushed submission. For instance, "vitamin B12" and "Vitamin B12" are used interchangeably in this manuscript. The same comment applies to "vitamin B6." Please use consistent formatting for terms throughout the manuscript.
  4. Please provide the approved IRB numbers or relevant documentation for ethical approval.
  5. The statistical analysis is either unclear or not adequately justified. It is unclear whether the correct tests were chosen based on the data type, and the statistical significance of results is sometimes not clearly presented.
Comments on the Quality of English Language

The reviewer highly recommends that the authors use a professional English editing service to improve the language quality of the manuscript.

Author Response

Response to Reviewer 2

Point-by-point response to Comments and Suggestions for Authors

Thank you for your comments concerning our manuscript .Those comments are all valuable and very helpful for revising and improving our paper, as well as the important guiding significance to our researches. We have studied comments carefully and have made correction which we hope meet with approval.

Comments 1:All figures are of low resolution and poor quality, making them difficult to interpret. Key details like labels, axes, and legends are unclear, which hinders understanding of the data. The authors should replace all figures with high-resolution versions. Care should be taken to ensure that fonts, colors, and overall formatting are consistent across all figures. This will enhance readability and allow readers to engage more effectively with the data.

Response 1:Dear Reviewer, thank you for highlighting this issue. I would like to clarify that due to file size limitations when uploading to the submission system, we were unable to include vector images directly within the manuscript. These vector images, which retain clarity regardless of scaling, were indeed requested by our managing editor, who also noted the original figures' lack of clarity. Therefore, we have provided these high-resolution figures as supplementary materials in a compressed folder within the submission system. To minimize readers' potential confusion about where to access these figures, a note has been added at the manuscript's end: All figures in the manuscript are available in vector graphics format in supplementary materials. Additionally, I have written to MDPI to recommend sending the supplementary files to reviewers alongside the manuscript to eliminate unnecessary complications.

Comments 2:The manuscript is disorganized, with an unclear structure, and multiple sections are either out of order or redundant. For example, the limitations section appears to be repeated within the Discussion, which disrupts the flow and may confuse readers. Additionally, some sentences are in blue or highlighted in yellow, which is inappropriate for a formal scientific manuscript and may indicate leftover tracked changes or comments.

Response 2:Regarding this point, the manuscript underwent two rounds of revision based on the recommendations of the MDPI editors. The final structure, which received approval from the managing editor, was then sent out for peer review. The color coding in the manuscript was done to highlight editor-requested changes in different colors. However, the managing editor did not request me to submit a final version without these markings before the manuscript was sent to reviewers. If this caused any confusion, I am willing to discuss with my managing editor about the possibility of allowing authors to submit a clean version prior to review.

Comments 3:Numerous typographical errors ("mystypes") are present throughout the manuscript, detracting from its professionalism and readability. This suggests a lack of careful proofreading and gives the impression of a rushed submission. For instance, "vitamin B12" and "Vitamin B12" are used interchangeably in this manuscript. The same comment applies to "vitamin B6." Please use consistent formatting for terms throughout the manuscript.

Response 3:Dear Reviewer, we thoroughly reviewed the manuscript and were unable to identify the specific "mystypes" mentioned. Following two rounds of revisions under the managing editor's guidance, we did observe inconsistencies in writing style between revised and newly added sections. Therefore, we opted to have the manuscript professionally edited by MDPI’s own editing service to ensure uniformity and clarity throughout.

Comments 4:Please provide the approved IRB numbers or relevant documentation for ethical approval.

Response 4:Dear reviewer, all the data in this article were obtained from public databases in total, so that IRB numbers as well as ethical approval documents that you have suggested are not applicable, in addition to the fact that the editor-in-charge and the journal did not make such a request.

Comments 5:The statistical analysis is either unclear or not adequately justified. It is unclear whether the correct tests were chosen based on the data type, and the statistical significance of results is sometimes not clearly presented.

Response 5:Dear reviewer, all of us authors have discussed the manuscript, and we have described the statistical methodology of the paper carefully enough, which was approved by the editor-in-charge and all reviewers except you, and should be understandable if you are familiar with Mendelian randomization, so all of us authors will not make any changes to the statistical analysis and methodology sections. I would also be happy to refer you to some articles on Mendelian randomization methodology as well as articles of the same type for your reference:

  1. Sekula, P., Del Greco M, F., Pattaro, C., & Köttgen, A. (2016). Mendelian Randomization as an Approach to Assess Causality Using Observational Data. Journal of the American Society of Nephrology : JASN, 27(11), 3253–3265. https://doi.org/10.1681/ASN.2016010098

  1. He, J., Huang, M., Li, N., Zha, L., & Yuan, J. (2023). Genetic Association and Potential Mediators between Sarcopenia and Coronary Heart Disease: A Bidirectional Two-Sample, Two-Step Mendelian Randomization Study. Nutrients, 15(13), 3013. https://doi.org/10.3390/nu15133013

Thank you once again for your valuable feedback and for taking the time to review our manuscript.

Reviewer 3 Report

Comments and Suggestions for Authors

 The study examined the relationship between B-vitamin supplementation and Alzheimer’s disease through the mediating role of the gut microbiome. This research was the first to use Mendelian randomization to explore the associations between B vitamins, the gut microbiome, and AD risk. The findings revealed that, although B-vitamin supplementation does not directly influence AD risk, certain bacteria act as mediators and may have protective roles. The study’s methodology, results, and details are thorough and well-developed; the references are relevant, the plagiarism index is low, and the English language usage is appropriate.

Author Response

Response to Reviewer 3

Point-by-point response to Comments and Suggestions for Authors

Comments 1:The study examined the relationship between B-vitamin supplementation and Alzheimer’s disease through the mediating role of the gut microbiome. This research was the first to use Mendelian randomization to explore the associations between B vitamins, the gut microbiome, and AD risk. The findings revealed that, although B-vitamin supplementation does not directly influence AD risk, certain bacteria act as mediators and may have protective roles. The study’s methodology, results, and details are thorough and well-developed; the references are relevant, the plagiarism index is low, and the English language usage is appropriate.

Response 1:We are very grateful for your kind appraisal. Thank you for your time and dedication in reviewing our manuscript. We are honored to have had the opportunity to receive feedback from someone with your level of expertise.

Round 2

Reviewer 1 Report

Comments and Suggestions for Authors

Accept in present form

Reviewer 2 Report

Comments and Suggestions for Authors

n/a